

# Beyond Perrault's experiments:
# Repeatability, didactics and complexity

Stefano Barontini[1] and Matteo Settura[2]

[1]DICATAM, Università degli Studi di Brescia, Brescia, Italy
[2]FISPPA, Università degli Studi di Padova, Padova, Italy

**Correspondence:** Stefano Barontini (stefano.barontini@unibs.it) & Matteo Settura (matteosettura@gmail.com)

**Abstract.** The studies conducted in the second half of the Sixteenth Century were crucial both for the birth of the modern hydrological science and for the modern epistemology. Thanks to quantitative observations and to the new experiment–based scientific approach, the Sun was about to be fully recognized as the engine of the hydrological cycle, instead of an endogenous engine within the Earth, originally conjectured by Aristotle to explain the water supply of great rivers. In this context of great

vitality and rapid cultural changes, two works were published on the origin of springs.

In 1663 Gaspar Schott published the *Anatomia physico–hydrostatica fontium ac fluminum* (Physical–hydrostatic anatomy of springs and rivers), deeply rooted in the Aristotelian epistemology and based on the ancient humanistic and dialectical method. In this book, Schott cited some observations recently made about the capability of the water to rise within a soil column. Pierre Perrault referred to these observations as a starting point to design the experiments reported in his classical opus *De l'origine*

*des fontaines* (On the origin of springs), published in 1674. With this approach Perrault places himself in the perspective of a deeply renewed epistemology: only a decade passed between the publication of Schott's opus and Perrault's one, but their perspective is radically different.

At the same time, the questions posed on the hydrological cycle and on the soil hydrology, which are hardly reproducible by means of a controlled laboratory model, severely tested the modern scientific approach at its beginning. Perrault seems to

be aware of such difficulties and, after discussing the results of the experiments, he moves the point of view from the scale of the laboratory model, to that of the catchment. This choice makes him a progenitor of the contemporary epistemology of complexity, which is framed both on laboratory analysis and on cases study. Thus even if Perrault's conclusions went in the direction of the ancient opinion, his work is not only seminal for hydrology, but also it helps to enlighten some intricate features of scientific revolution.

Aiming at contributing to understand the importance of Perrault's opus, we will discuss his epistemological relevance through the lens of the repeatability of the experiments, of the intriguing didactic aspects which, starting from his experiments, arise for modern teaching of hydrology, and of his attitude to face the complexity of the hydrological processes.





## 1 Introduction

By the half of the XVII century, Hydrology was at a crucial passage of its evolution, because the Sun was about to be fully recognized as the engine of the hydrological cycle, instead of an endogenous one within the Earth, originally conjectured by Aristotle to explain the water supply of great rivers. Anyway, in order to become a modern science, Hydrology needed two

steps, that are the concept of scientific experiment and the assessment of the quantitative evaluation of the water cycle. In his classical opus *De l'origine des fontaines* (On the origin of springs, Figure 1), Pierre Perrault (1611–1680) took both these steps. He performed experiments and collected many observations both to assess the water balance at the basin scale and to understand the water movement in the upper soil layers (Perrault, 1674, 1678, 1967, 1986). His work therefore attests that the birth of the modern hydrological science is strictly linked to the Scientific Revolution of the XVII century. Indeed, Perrault

was able to achieve the passage to scientific hydrology because he had a new, critical attitude towards tradition, which was also born by scientist–philosophers like Descartes and Pascal (Tixeront, 1974).

Perrault's seminal opus may be framed in the context of the *querelle des anciens et des modernes* (controversy between the Ancients and the Moderns), a spirited debate that took place in France during the second half of the XVII century and had profound resonances all–over Europe (Fumaroli, 2001). Although the controversy emerged in a literary milieu, it rapidly

spread in the scientific field, so as to intertwine the debates of the nascent Scientific Revolution. Pierre Perrault was a cultured man of noble origin and he played a role in the controversy, as well as his brothers, who were prominent figures in various fields of politics and culture (Nace, 1974). For instance Perrault's brother Charles, a renowned poet of the age, was author of four volumes focused on the question of whether the primacy in art and knowledge should lie with the Ancients or with the Moderns (Perrault, 1688–1697).

The epistemological meaning of this opposition was immediately evident for the intellectuals of the age: at stake was the correct way of achieving the truth. Whereas the supporters of the Ancients defended the pre–eminence of the old principle of authority, the Moderns put forward the idea that knowledge must be grounded only on reason and experience. The *querelle* can thus be seen as the way French intellectuals of the *age d'or* dealt with the deep cultural transformations of their time. Indeed, the comparison between the modern knowledge and skills and those inherited from Antiquity represented for them an

effort to become aware of the historical and intellectual breakup they were living. Like his brother Charles, Pierre Perrault took side with the Moderns, applying the new experimental method to the problems of soil hydrology for the first time. Therefore, it is no surprise to find in his writings a strong reflection of the spirit of Scientific Revolution and a thorough awareness of its achievements. He appears fully conscious of the difference between his own approach and that of the predecessors and considers the experiments as the primal source of scientific knowledge:

I know all too well that it is to experiments that we owe the finest knowledge we now have concerning the things
of nature which had been unknown to Antiquity; and I base myself only on such principles (Perrault, 1967, p.11).

In the introductory *Avertissement* (Warning) of the *Origine* Perrault explains his method and circumscribes the field of his enquiry, stating that he will not present "great reasonings of Physics". Rather, he will speak "with a simplicity which can appeal to the most ordinary brains" (Perrault, 1967, p.9). In so doing, the author immediately distances himself from the Aristotelian




approach, according to which Natural Science was a "logical deductive structure derived from uncontestable basic statements or premises" (Applebaum, 2000, p.341). Perrault is not interested in discussing the primal causes of natural phenomena and the principles of Physics. He focuses only on the specific issue of hydrological cycle, providing sensate experiences to support his opinion: "my intention has been to speak only of that which is obviously perceived by the senses" (Perrault, 1967, p.10). Such

a declaration of intents is in–line with the *incipit* of René Descartes' seminal opus, the *Discours de la méthode* (Discourse on the method), a text that aimed to divulge the correct approach in "conducting one's reason and seeking truth in sciences", as stated in the title (Descartes, 1637, 2006). Descartes plainly rejects the traditional centrality of classical studies and, therefore, the authority of past philosophers. On the contrary, he claims that "good sense is the most evenly distributed thing in the world" and science is in principle accessible to anyone, provided that one proceeds step by step, "for it is not enough to possess a good

mind; the most important thing is to apply it correctly" (Descartes, 2006, p.5). According to Descartes, the method should not start from general principles derived by the philosophical heritage:

> As for the (. . . ) disciplines, in so far as they borrow their principles from philosophy, I concluded that nothing
> solid could have been built on such shaky foundations (Descartes, 2006, p.10).

On the contrary, the path of science can only consist in an analytically ordered succession of experiences, each one supported

by the constant check of reason. Thus, in this new perspective, reason does not dictate abstract principles, rather it plays an ordering and regulative role. At the same time, reason and experience are not opposite, for the latter provides the indispensable "matter" to any reasoning (Descartes, 2006, p. 20).

A similar centrality of experience can be found also in Pascal's *Préface pour un traité du vide* (Preface to the treatise on vacuum, written in 1651), which is another relevant reference to understand Perrault's position (Pascal, 1954, 1910). Here,

the French philosopher and scientist states that whereas authority is essential with regard to literary and historical issues, it is useless in the domain of matters that fall under our senses, such as the investigation of nature. In a few pages Pascal establishes the basis for the separation between humanistic and scientific attitude which still persists nowadays, yet with important differences. In fact, Pascal's separation between the so–called "two cultures" (Snow, 1998) is an argument against those academicians who wished to submit science to the dogmatic authority of Metaphysics and Theology. According to the

author, the investigation of nature is characterised by "freedom", as it is based on experience, whereas "dogmatism" is typical of literary, philosophical and theological studies. This position paved the way to the overcoming of Aristotelianism: as far as experience plays a crucial role, the Philosopher's word could not be the final word in the knowledge of nature. In his *Avertissement* Perrault shares this posture towards the past authors without falling into arrogance, as he claims that:

> It is not an extraordinary thing to examine the ideas of any author, it is permitted to the least disciples to ask the
most renowned Philosophers, in the very school where they have most credit, the reasons for the propositions that
> they have avanced (. . . ) (Perrault, 1967, p. 10).

Beyond the consonance of the epistemological attitude, it is also noteworthy that Pascal's role in the development of scientific hydrology was not merely speculative. He knew the work of Galilei and Torricelli, which shaped modern Mechanics and Hydraulics, and he tried in many different ways to reproduce the experiment performed in 1644 by Torricelli, to measure the





atmospheric pressure (Tixeront, 1974; Applebaum, 2000). His measurements and observations were important terms of comparison for the implementation of Perrault's experiments. Although Perrault criticises some of Pascal's assertions on specific issues (Perrault, 1967, p.150), the epistemological perspective and the method that underpins his work refers to the position of Pascal's *Préface*. On this basis, we can claim that Perrault is permeated with the critical attitude that should characterise a

*modern* scientist. In his critique of both ancient (e.g. Aristotle) and modern scholars (e.g. Nicolas Papin) he makes use of concepts drawn from the works of Bacon and Galilei, Descartes and Pascal. Thus the driving forces in the passage to quantitative hydrology are the overturning of the primacy of *auctoritas* and the centrality of experiments as primal source of knowledge. Having clarified the major cultural premises of Perrault's work, in Section 2 Perrault's opus will be compared with that of his contemporary Gaspar Schott, to point out the shift of paradigm he brought in soil hydrology. In the followings, Perrault's

experiments will be investigated through the lens of their repeatability (Sections 3 and 4), presenting also their possible didactic implications (Section 5). Finally Perrault's seminal contribution to the perception of complexity in the hydrological science will be enlightened (Section 6).

## 2   Comparison between Schott's and Perrault's opus

How deep was the paradigmatic renewal fostered by Perrault with his opus may be realized by comparing the *Origine* with

another opus, the *Anatomia physico–hydrostatica fontium ac fluminum* (Physical–hydrostatic anatomy of springs and rivers, Figure 2), published only ten years before by the Jesuit Gaspar Schott (Schott, 1663). Schott carefully reviews and discusses the opinions of ancient and modern authors about the origin of springs and rivers (see Table 1), finally stating his own conjecture, according to which the sea is the origin of most of the springs, and the water rises from the sea mostly due to the greater altitude of the sea, with respect to the mountains. The *Anatomia*, which was written in Latin – whereas the *Origine* was written

in French, thus being intended to be open to a broader audience –, is an *opiniones* (opinions) book, deeply rooted in the ancient humanistic and dialectical approach and far from the modern scientific method. A comparison between the form and structure of the two texts reveals further important differences, despite the short lapse of time which separates them.

As a start, it is noteworthy how Schott felt it necessary to focus on histories of ancient springs, drawn by classic literary sources, in which mythological and imaginary witnesses are mixed together with historical and geographical accounts (Schott,

1663, pp.12–63). In Schott's view the purpose of scientific description does not conflict with the recreational intent of literature and the stories handed down by the Ancients still represent for him the point of departure for intellectual investigation. Therefore it seems that he does not take into account the distinction between humanistic and scientific discourse proposed by Pascal. In contrast, Perrault does not take in any consideration the mythological and literary tradition about springs, showing a deep awareness of the above–mentioned separation between the "two cultures". However, both the authors provide an extensive and

detailed discussion of past scholars' opinions about the origin and nature of springs and rivers. Here the difference between Schott and Perrault lies both in the way the discussion is conducted and in the authors which are cited (or not) by them. It can be assessed in general terms that Schott's text does not present a clear demarcation between the exposition and the critique. In addition, his critiques appear often to be grounded on philosophical considerations and on the appeal to authority arguments.




In particular, he dedicates a detailed examination to the Holy Scriptures, which are regarded to be reliable also as scientific treatise (Schott, 1663, pp.88–102).

By contrast, Perrault's opus provides, for each author discussed, a clear separation between exposition and critique and, most important, he motivates his objections on the basis of the experimental results which are presented in the Second Part of the work. He does not consider the Holy Scriptures as a pertinent source in the context of his work. This choice represents another indication of his critical attitude towards the principle of authority. Furthermore, Perrault discusses a set of authors that he defines as supporters of the *Opinion commune* (Common opinion) (Perrault, 1674, p.150 ff.). These authors are Vitruvius, Bernard Palissy, Paul Gassendi and père Jean–François. It is noteworthy that Vitruvius' capital opus *De Architectura* (On architecture) was translated into French by Perrault's brother Claude, one of the major architects of the time, in 1673 (Vitruvius and Perrault, 1673). All these authors share an interest in the practical application of hydrological knowledge, and all of them are absent from Schott's treatise. This discrepancy reveals the different epistemological positions held by the two authors: whereas Schott still considers his work as an essentially intellectual activity and as definitely distant from the practical problems of technical application, Perrault regards hydrology as strictly connected with the sphere of technique and engineering.

It is in our opinion no coincidence that Schott did not refer to Palissy's work. Indeed Bernard Palissy was not a scholar, but an artist ceramist. His strong attitude for the technical investigation is reflected also in the *Discours admirable* (The admirable discourses, 1580), where he imagines a polemical dialogue between the personifications of Theory and Practice (Palissy, 1580, 1957). The triumph of the latter signifies the superiority of the "vulgar methods" over the scholastic and speculative approach of the traditional philosophy of nature. Although Perrault argues against Palissy and the Common opinion, he agrees with him that practice could teach more than books about nature. However, Palissy's approach still reflects a Renaissance idea of experience, related to the craft work, in which the artist craftsman subjectively learns from his experience and on the basis of his own sensibility. By contrast, the Scientific Revolution broke the tie between art and knowledge of nature, the latter deriving from series of ordered experiences whose objectivity is guaranteed by the repeatability of measures. Perrault does not only affirm the superiority of experience over authority, he also *performs* an experiment and, most importantly, provides an experimental report whose accuracy fits the standards of the modern scientific practice. Differently from Palissy, Perrault felt the necessity of submitting concrete and checkable results in order to support his thesis.

Getting back to the comparison between Perrault and Schott, the crucial difference still lies in the fact that the former supports his theses with the report of an experimental activity. In fact, after discussing previous authors' opinions (Perrault, 1674, pp.8–146), Perrault enunciates his idea (Perrault, 1674, pp.148-150, Figure 3), according to which water cannot easily infiltrate, flow through the soil, and reach great depths, but it is retained in the superficial layers, and directly contributes to the evaporation and marginally to refill the surface springs. He submits his theses in contrast with the aforementioned Common opinion (Perrault, 1674, pp.150–152), according to which water could penetrate the soil until it reaches a layer of *terre grasse* (translated as "fatty earth" in Perrault, 1967, p.75) or rocks, that would laterally deflect its pathway until it reaches the surface, so as to form a spring. Perrault disputes this view on the basis of two major objections. First, he calls into question that water could really penetrate the soil to depth; second, he doubts that it could spontaneously outcrop (Perrault, 1678, pp.153–154). To prove the validity of his objections, Perrault designed and implemented an experimental activity, the report of which can be





found in the subsequent pages (Perrault, 1674, pp.154–160, see Section 3). He concluded that the water alone has no tendency to deeply penetrate the soil nor to spontaneously outcrop to form springs, thus corroborating his theses (Perrault, 1674, p.160 ff.). The remaining part of the *Origine* contains a general discussion of the hydrological cycle, in which Perrault shows some evidence for the fact that about one–sixth of the precipitation is sufficient to provide the water flowing in the rivers (Perrault,

1674, p.204 ff.). Finally he admits that small springs could be affected by precipitations and snow melting, but he points out that the general circulation of water in soil is necessarily supported by a process of evaporation and condensation within the Earth, in line with the model described by Descartes. It is worth noting that the external circulation of the hydrological cycle was originally recognized by some Presocratics (Brutsaert, 2005, p.562), but it was soon coupled by Aristotle with the idea of the presence of an internal evaporation and condensation process, taking place in the Earth's depth (Aristotle, 1952, Book 1).

This dualism of the hydrological circulation underlay all the hydrological culture since Aristotle to Perrault and beyond (see Biswas, 1970; Duffy, 2017, for an overview).

Finally in his book, Schott reports about some observations recently made by Maignan about the capability of the water to rise within a soil column, to which Perrault will refer as a starting point to design his experiments:

> Fill, with dry sand, a glass pipe opened on both sides, then cover the lower mouth with a cloth so that the sand
cannot pour out; and dip the covered mouth into water only for the altitude of a transverse finger: you will observe that water rises a little and all the sand is wetted, up to the height of three palms above the surface of the water [which is] in a vessel (Schott, 1663, p.254, our translation).

Despite Schott and Maignan provide these few approximative measures, their perspective is focused on the qualitative behaviour of the water within the sand. The form of Schott's argumentation is still shaped on the model of the scholastic dialec-
tical disputation between conflicting opinions. Even when he introduces elements derived from experience, they seems casual remarks, which are not crucial in order to address the controversies. On the opposite, Perrault designs experiences in order to support his hypotheses and he places the report of his experiments at a turning point of the book, after which the conclusions are thoroughly and quantitatively discussed. With this approach he places himself in the perspective of a novel epistemology which reflects the major aspects of Scientific Revolution: the expunction from the discourse of the elements derived from mythical,
literary and religious tradition, and the attention paid to practical and technical applications are combined with the passage to the quantitative evaluation of the phenomena, which makes possible the design of crucial experiments, i.e. of experiences which are diriment and which can be repeated.

## 3  Perrault's experiments

At p.154 of his opus, Perrault (1674) introduces his experimental activity. We reproduce here his note in the English translation:

> Before beginning the discussion of these two difficulties, I wish to describe here an experiment I have made, which may shed a little light on what we have to say (Perrault, 1967, p.78).





His experience is effectually grouped into four experiments that he performed with a leaden pipe, filled with various soils and in various imbibition conditions. Before presenting the repetitions of his experiments, in the following we will briefly summarize what is reported by Perrault. The measurements given by Perrault are converted into the International System according to the *Toise de Châtelet* system, which entered into force in 1668 (Blamont, 2001, see Table 2).

During the first experiment Perrault took a leaden pipe, $65\,\text{cm}$ long and with a diameter of $4.5\,\text{cm}$. He closed it at one side with a cloth and filled it with river sand, sieved with a coarse sieve. He put the pipe vertically and dipped it into a wide water–filled vessel, leaving it in this position for one day. After 24 hours, as he reported, he found that the water rose within the soil for $48.7\,\text{cm}$. In this point it is not clear how he performed this observation, as elsewhere (at the end of the description of the first experiment) he reports that in order to appreciate how the soil was wetted, he needed to shake the pipe to let the soil slide out

from the pipe. Maybe he reports here a measurement that he performed at the end of the experiment, or he extracted the sand, he made the measurement and than filled the pipe again to proceed with new measurements. As a first conclusion he reports that he was surprised at such a rise of the water within the column. Then he wanted to check whether a spring could spontaneously outcrop above the water surface. He therefore made a hole about $5.4\,\text{cm}$ above the water surface, with a diameter between $1.6$ and $1.8\,\text{cm}$, and in order to restore the medium continuity between inside and outside the pipe, he added a $5.4\,\text{cm}$ long spout,

sloping toward the vessel, without reaching the water surface. The spout was covered by a grey paper, to evidence the wetting, and by a thin sand layer. He commented that in case a spring had onset, then perpetual motion would have established, but he only observed that the sand and the paper were wetted, without producing any drop of water. It is worth noting that this case was already recognized by Schott (1663) in his Figure XI (see Figure 4 in this paper), according to which a spring might onset only if the extreme of the spout (a cloth, in Schott's example) was below the water surface in the vessel. Afterwards Perrault

suspended the pipe over an empty vessel in order to check whether the soil allowed some water to drain, but also in this case no water was collected within half a day. Then he twice added a certain amount of water (which is not specified): the day after the first imbibition he collected $\frac{3}{4}$ of the added water, and the day after the second one he collected all the added water. Finally he wanted to check the water content of the soil and shook the pipe in order to let the soil slide out, thus recognizing that the lower part of the soil was thoroughly wetted as "very wet mortar" (Perrault, 1967, p.79). The second experiment was the repetition

of the first one with three different soils (details are reported in Table 3). Perrault reports that he made the same observations with different rise ($10\,\text{pouces} \simeq 27.1\,\text{cm}$ for the sand with gravel and for the sieved sandstone, and $18\,\text{pouces} \simeq 48.7\,\text{cm}$ for the crumbled loam). Perrault's measurements are reported in Table 4 together with our repetitions.

Perrault performed a third experiment to check whether the soil could let the water during the precipitation reach a layer of impervious soil or rock, where it could stop and rise to form a spring. In order to do so, he took the pipe and filled with dry

and sieved loam, compacting it with a rod. He suspended the pipe over an empty vessel and poured many times the amount of water contained in a glass vial, filled up to the spout, and with the same volume of a medium ball of the *jeu de paulme* (today *jeu de paume*, palm–game; Perrault, 1967, p.80). The practice of measuring liquid volumes with reference to the balls of the *jeu de paulme* was quite common in XVII and XVIII century France. Yet it was not possible to find a reference for the actual dimension of a medium ball of the *jeu de paulme* at Perrault's age, therefore in order to repeat the experiment its volume was

deduced from the notes reported by Perrault in the description of the third experiment. In fact he slowly poured the water on the





soil three times without recovering any water at the bottom. Afterwards he added a fourth vial and recovered $\frac{1}{3}$ of its volume and the flux stopped for at least 18 hours. He added two more vials, collecting all their volume. He waited three days and added a seventh vial, collecting $\frac{3}{4}$ of the volume (he observed that the surface of the soil was drier than before, because some water evaporated), and an eighth one, collecting all the volume. Then he extracted the soil from the pipe and observed that only the

5 lowest $48.7\,\mathrm{cm}$ were very wet, so that he concluded that the soil may leak the water only if it is thoroughly wetted before, and in order to do so they are required $3 + \frac{1}{2}$ vials which correspond to the third part of the volume of the wet soil. Even if there is a little incongruence between the data (after the first imbibition $3 + \frac{2}{3}$ vials were stored in the soil), this information is valuable because it let us to conjecture that the volume of Perrault's vial was about $74\,\mathrm{cm}^3$ (Barontini et al., 2013).

In the fourth reported experiment Perrault aimed at testing whether salty water, rising by capillarity within the soil, would

10 have maintained its salt content or it would have left it in the sand. As in fact the spring water is commonly fresh, this was considered a key aspect to understand the circulation of water within the Earth. Perrault's report is not detailed in this case, but he only notes that, after the water rose in the sand column for $48.7\,\mathrm{cm}$, the sand was salty both at the top and at the bottom, and he guessed that if the sand at the top was less salted then that at the bottom, it should be a consequence of the fact that the soil was less wet at the top than at the bottom.

15 About two pages (Perrault, 1674, p.160–162) are devoted to summarize the conclusions that Perrault made after the experiments. It is worth noting that the first conclusion starts from the experiment to give a general epistemological comment on the attitude of the authors at deriving general conclusions from limited observations:

> First I know that Magnanus' opinion [Maignan's opinion] is not acceptable, and that it is based on a fact which is
> only half true: upon which I note in passing that what I have said elsewhere is really correct that most of those who
20 > make experiments on discoveries of which they wish to pass as the Authors, will look at them only in the aspect
> that serves their ends, as did Magnanus who only wanted to prove the absorption of water by the Earth (Perrault,
> 1967, p.80).

The other conclusions directly refers to the experiment. They may be summarized saying that once a soil is wet, it tends to retain water and not to give it back, and this is true also when the soil is so wetted to become permeable to water. In that case

25 in fact as soon as the water addition stops, the percolation rapidly diminishes and at the end the soil still retains the amount it needed to become permeable. Perrault then comes back to the discussion of the two difficulties he found regarding the Common opinion (Perrault, 1674, pp.162–183) and concludes that:

> After what I have just observed concerning the two difficulties which I have found in the Common opinion, I think
> it must be agreed that the penetration of the Earth by rain water cannot be achieved according to this opinion; and
30 > the rains to which it attributes the origin of springs not being enough to produce them, and still less to cause them
> to flow continually, it is proper to reject this view (Perrault, 1967, p.89).





## 4 Repetition of Perrault's experiments

The great detail of Perrault's experimental report posed us two questions: Are Perrault's experiments really qualitatively and quantitatively repeatable in the laboratory? Is it possible, on the basis of Perrault's data, to conjecture which could be the hydrological properties of his soils? The questions were addressed both in the laboratory (Berta and Barontini, in preparation)

and with numerical simulations (Barontini et al., 2013). Some aspects of the numerical simulations will be recalled in the next sections, here the main laboratory procedures and results will be summarized. The experimental details and data are fully reported by Berta and Barontini (in preparation).

    A polycarbonate pipe of $4.5\,\mathrm{cm}$ of diameter, cut at the length of $65\,\mathrm{cm}$, mimicked Perrault's leaden pipe. A punched metal plate and a cloth of nonwoven fabric were used to hold the soil and drain water when the column was held in a vertical position

(Figure 5). Three quarry sands, commonly used in building works, were chosen to repeat the experiments. In the followings the soils will be referred to as A, B and C, according to the order in which they were used for the repetitions. The grain size distribution curves were determined by dry sieving, as the fine soil component was negligible. The soils were classified as coarse–to–medium sand (A, C) and medium–to–fine sand (B). The soils were poorly graded, with uniformity coefficient $U = d_{60}/d_{10}$ about 2.1 for A and B, and about 2.2 for C. The hydraulic conductivity at soil saturation $K_s$ was hardly estimable

by means of the laboratory falling–head permeameter, because expected values were greater then $10^{-5}\,\mathrm{m/s}$, therefore it was estimated by means of Kozeny and Carman's equation (see Eq. A1 in the Appendix). The soil–water retention relationships were measured by means of Richards' pressure plates for pressure values $\Psi \leq -10\,\mathrm{kPa}$ (Richards and Fireman, 1943), and by means of an hydrostatic method for $-3\,\mathrm{kPa} \leq \Psi \leq -1\,\mathrm{kPa}$. The couples $(w, \Psi)$, being $w$ the gravimetric water content, were interpolated by means of Brooks & Corey's retention curve (A4) and van Genuchten's retention curve (A5), both with or

without the usual constrain (A6) on $m$ and $n$. An example of the obtained soil–water retention relationships is represented in Figure 6 for the medium–to–fine sand (soil B). Values at $-3\,\mathrm{kPa}$ were too sparse and were not used for the regressions.

    In Figure 7 and in Table 4 and 5 the results of the repetitions of the first experiment are reported. The unitary amount of water poured on the soils was in this case $74\,\mathrm{cm}^3$ for all the soils, as determined by Barontini et al. (2013). All the soils show a qualitative and (at some extent) a quantitative agreement, both during the spontaneous imbibition from the bottom, and during

the forced imbibition from the top. In Table 4 the measured capillary rises, both those measured after one day and the limiting values, are compared with data reported by Perrault and with the capillary rise $h_{cr}\,(\mathrm{cm})$ determined by Polubarinova–Kochina's relationship (Polubarinova-Kochina, 1962):

$$h_{cr} = \frac{0.45}{d_{10}} \frac{1 - \phi}{\phi}, \qquad (1)$$

where $d_{10}\,(\mathrm{cm})$ is the equivalent diameter which retains $90\%$ of the soil weight, and $\phi$ is the soil porosity. The good agreement

between the measured capillary rise and $h_{cr}$ allows us to conjecture that Perrault's soils almost reached their asymptotic rise. Thus, accounting for a porosity $\phi = \frac{1}{3}$ (because Perrault estimated the amount of water in the soil as $\frac{1}{3}$ of the wetted soil volume) is it possible to estimate the competent characteristic diameter $d_{10}$ (values are reported in Table 4).

    To characterize the hydraulic conductivity at soil saturation is less immediate because the information given by Perrault about the percolation times are incomplete. If we consider that at least some of the soils used by Perrault were sieved in order





to remove the coarse fraction, we hypothesize that the characteristic diameter $d_{10}$ determined by means of Polubarinova–Kochina's equation (1) is an approximation of the characteristic diameter $\tilde{d}$ (Eq. A3) to determine $K_s$ by means of Kozeny and Carman's equation (A1). Comparing $d_{10}$ and $\tilde{d}$ as determined for soils A, B, C (see Table 4) this conjecture may be considered realistic.

Another interesting comparison between Perrault's experiments and their repetition is done with regard to the amount of water that was added and that percolated. In Table 5 the imbibition and percolation volumes are represented. Values are expressed as number of vials and in this case the volume of a vial was assumed equal to $74 \, \mathrm{cm}^3$. Data are qualitatively in agreement, and although they numerically differ, it emerges that Perrault's data are complete and quantitatively comparable. The same data are presented in Figures 8 and 9 and in Table 6 for the third experiment. It is worth recalling that in this case the

volume of $1 \, \mathrm{vial}$ was chosen with reference to the capability of the soil of retaining water. In fact in his conclusions regarding the experimental activity, Perrault stressed his point of view about the proneness of a soil to allow water to drain, and he considered that this may happen when the soil is wetted with an amount of water which correspond to $\frac{1}{3}$ of the wetted volume of soil at equilibrium. In the conclusions in fact he states:

> . . . that in order to wet earth and make it ready for penetration as much water as one third the height and thickness
15       of the earth is necessary (Perrault, 1967, p.81).

As he stated that this amount of water corresponded, in the investigated case, to $3 + \frac{1}{2}$ vials, we obtained the unitary volume $V_i$ of a reference vial, for each soil, dividing the volume of water stored at equilibrium $V_w$ by $3 + \frac{1}{2}$. The determined unitary volumes are $V_i = 31.3, 50.5, 41.0 \, \mathrm{cm}^3$ for the soil A, B and C, respectively. Also in this case we observe a good qualitative correspondence between Perrault's experiments and the repetitions, but what is important to stress is the fact that data provided

by Perrault allow us to make a robust quantitative comparison.

    With regard to the repeatability of Perrault's experiments we may therefore conclude that they are almost completely repeatable, in a qualitative sense. If we assume that a quantitative repetition should not be intended at obtaining the same results obtained by Perrault, but at defining results which may be compared with Perrault's one on a quantitative basis, we may also say that Perrault's experiments are also quantitatively repeatable. The detail in fact with which Perrault describes his experi-

mental activity allows us $(a)$ to clearly identify most of the quantities that he measured, $(b)$ to make realistic conjectures for the quantities about which the *Origine* is not precise (i.e. the unitary amount of water added with one vial) and $(c)$ to recognize the lack of information (i.e. the time running between the beginning of the imbibition, the beginning of drainage and the end of drainage). In the next section some conclusions will be made also with regard to the second question posed at the beginning of this section.

## 30   5   Didactics of Perrault's experiments

Perrault's opus and experiments can be used in several didactic ways, meeting the objectives of at least the first three elements of the Dublin descriptors, i.e. knowledge and understanding, applying knowledge and understanding, making judgements (Bologna Working Group on Qualifications Frameworks, 2005). As a general preliminary aspect, facing the *Origine* and the





reported descriptions of the hydrological cycle induces the students to reflect on the long scientific path, which was needed to move from initial observations and conjectures to the modern description of the phenomena. It stimulates also an exercise of displacement from consolidated understandings (which may be considered our *Opinion Commune*) and the exploration of alternative descriptions.

Focusing with more detail on soil hydrology, Perrault's experiments offer a wide range of didactic cases, useful to teach and exemplify many aspects of the subject, so that it was used as an underlying cross theme of a recently published didactic book (Barontini, 2017). The description given by Perrault of the first and second experiments in fact offers the opportunity of teaching the hydrostatics of the soil–water, both above a groundwater table and above a capillary barrier, and of presenting the capillary fringe and the soil–water retention curves. The soil resistance to be crossed by the water, reported in the third experiment, is

related to the action of the capillary barrier exerted by the cloth at the bottom of the soil column. It stresses the importance of locally reaching the soil saturation so as to have the continuity of the generalised piezometric potential $\Phi = \Psi + z$ at the soil bottom ($z$ being the elevation), before percolation will start. This issue is directly related to Perrault's observation regarding the soil capability of retaining water and about the difficulties of infiltrated precipitations to percolate. It therefore leads the students to understand the importance of soil layering on field capacity. The soil column in fact was suspended and the soil had

greater field capacity than if it was in the field, so that Perrault was induced to think that precipitations have great difficulties to percolate toward deep soil layers. The fourth experiment, even if minimally described, also allows the hydrostatics of an unsaturated soil with solutes to be introduced.

    The experiments allow a great variety of boundary conditions to be explored, including the condition of known pressure potential (bottom boundary of the first experiment), of null flux (top boundary of the first experiment) and of seepage (bottom

boundary of the third experiment), but not the condition of known flux at the top boundary, because Perrault does not report how long the imbibition lasted during the third experiment. This, together with the other incomplete or omitted data, trains the students with the problem of recognizing the goodness of the information they may find. Moreover trying to interpret and understand ancient observations with modern conceptual frameworks is an interesting exercise to test and apply the gained knowledge.

Finally students may face the second question previously introduced, regarding our capability of inferring the hydrological properties of Perrault's soils, on the basis of the data reported. In the previous section the problem of the conductivity at soil saturation was introduced. Now we will focus on the soil–water retention relationships.

    If Perrault's soils are regarded to as $\Delta$–soils, the capillary rise $h_{cr}$ defines also their soil–water retention relationship as:

$$
\begin{cases}
s = 1 & -h_{cr} \leq \Psi \leq 0 \\
s = 0 & \Psi < -h_{cr}
\end{cases}
. \tag{2}
$$

$\Delta$–shaped soil–water retention relationships may be considered a first approximation of the hydrological characteristics of Perrault's soils, but other soil–water retention curves are compatible with the information given by Perrault. As an example which is effective for the soil of the first experiment, the hypothesis is introduced that the capillary rise of $48.7\,\mathrm{cm}$, observed by Perrault, is between the scale factor $\Psi_1$ and the inflexion point $\Psi_f$ of van Genuchten's soil–water retention relationship





(Barontini et al., 2013). Therefore, one can fix either $\phi < \frac{1}{3}$ and $\Psi_1 = 48.7\,\mathrm{cm}$, or $\phi < \frac{1}{3}$ and $\Psi_f = 48.7\,\mathrm{cm}$, such that the remaining parameters of the soil–water retention relationships may be determined, provided that at equilibrium the volume of water stored in the soil $V_w$ is:

$$V_w = A \int_0^{65\,\mathrm{cm}} \theta(x)\mathrm{d}x \equiv 3.5 \cdot 74\,\mathrm{cm}^3. \tag{3}$$

In Eq. (3) $\theta$ is the volumetric soil–water content. An analogous hypothesis may be introduced with respect to Brooks and Corey's bubbling pressure $\Psi_b$ (Barontini, 2017), and in both cases the soil will approximate a $\Delta$–soil at increasing $n$ or $\lambda$, and consequently at increasing the porosity $\phi$. The $\Delta$–soil will therefore be the limit case for $\phi = \frac{1}{3}$.

Anyway both these hypotheses, according to Perrault's epistemological approach, may be now tested with the soils used for the repetitions of Perrault's experiments. The results are reported in Figure 10 in which the observed capillary rises are plotted

together with Polubarinova–Kochina $h_{cr}$ and with the estimated values of $\Psi_b$, $\Psi_1$ and $\Psi_f$. The hypotheses are not rigorously satisfied, but the capillary rises are comparable with $\Psi_b$, $\Psi_1$ and $\Psi_f$ and this indicates that to some extents the hypotheses may lead to a realistic approximation of Perrault's soil constitutive laws. The results might be better, if the experimental soil–water retention curves were measured with more detail near to saturation, but what is important for teaching purposes at this point is that the comparison between the hypotheses and the experimental evidence induces the students to critically analyse their

*modus operandi* and to understand which are the strong constraints of the epistemological approach based on the experimental practice.

With regard to the second question posed at the beginning of Section 4, we may therefore conclude that it is not possible to define a set of hydrological properties which completely describes one of Perrault's soils; yet it is possible to make a set of reasonable hypotheses, which allow a range of values to be defined that are realistic for the hydrological properties of the

described soils.

## 6   Perrault and the complexity

Perrault's discussion to confute the *Opinion commune* (Perrault, 1674, pp.162–183) is conducted, at the same time, by using philosophical conjecture, by claiming the importance of his experimental activity:

What the earth drinks, this Philosopher [Seneca] says, is little (. . . ); but I add to this reasoning experiments made

daily on this penetration of the earth (Perrault, 1967, p.81),

and above all by showing a great attention to field phenomena:

To understand this [the second difficulty with the Common opinion] completely, one must picture and discuss in detail, how the penetration of the Earth according to the Common opinion can be accomplished. The water that falls on the Earth, begins by wetting the parts of the earth or sand nearest to it; then it wets other, more distant

parts, then others, always going downward. and wetting the Earth in all its parts one after the other (Perrault, 1967, p.86).





This attention permeates all the following pages and makes him move the focus from the laboratory to the nature.

Here Perrault, according with his refusal of the principle of authority, challenges his perspective with the famous Aristotle's assertion, according to which the total mass flowed by springs or in rivers during one year would be greater than the whole Earth mass:

5     The name of its Author, the number of rivers on the Earth and the length of a year, are things that strike the imagination so strongly that it is hard not to be won over to such a likely idea, and such a hard one to argue about. But without being amazed, let us try to consider this objection, and without distrusting our strength overmuch, let us try to find a solution for it (Perrault, 1967, p.90).

Discussing Aristotle's statement, Perrault concludes that the total amount of water flowing in the rivers is great, but much less 10 than Aristotle says. In order to reinforce his conjectures, Perrault recognizes the necessity of collecting measurements against those who do not agree on the fact that the precipitation volume is greater than the runoff volume:

(. . . ) but as these reasons bear only on the destruction of the opposing opinion, some attempt must be made to give other reasons that might support the one that I maintain and show that rain waters are enough to cause the flow of springs and rivers for a whole year. (. . . ) I shall try nevertheless by making rough estimates of the volume of rains 15     and of that of the flow of rivers, to give some basis for judgement on the opinion that I uphold, and to arrive at such judgement (Perrault, 1967, p.95).

As a first step he clearly establishes the methods of measurement he will employ in his observations:

It is necessary above all to reach an agreement on ways of measuring these two kinds of water (Perrault, 1967, p.95).

20 Then he reports the results of his measurements of the precipitation and runoff made between 1668 and 1671 in the Seine catchment, thus concluding that:

only about one–sixth part of the rain and snow water that falls, is therefore needed to cause this river to flow continually for one year (Perrault, 1967, p.97).

Which is the value given by Perrault to this conclusion? He seems to be aware that the measurements are imprecise and not 25 complete, but in this context, rather than at providing exact results, he aims at showing a research path that will lead to the birth of the modern hydrology:

I know very well that this deduction has no certainty: but who could give one that would be certain? Nevertheless whatever this one may be, I believe it is more satisfactory than a simple negative like Aristotle's and that of those who maintain, without knowing why, that it does not rain enough to supply the flow of rivers. However that may 30     be, until someone makes more exact observations, by which he proves the contrary of what I have advanced, I shall hold to my view, and be content with the feeble light shed by the observations I have made, being unable to have a stronger one (Perrault, 1967, p.97).





The estimation of the water balance led Perrault to recognize that the precipitation provides enough water to guarantee the flow in the rivers, so that he should have admitted that both rivers and springs are caused by rain and snow fall. Yet Perrault takes his experiment as a milestone, and he rejects the possibility that the precipitation may percolate so deeply to reach the impermeable layer and to origin both the groundwater table and the springs. Therefore, in order to explain the origin of springs,

with particular reference to the mountain springs, Perrault surprisingly comes back to the undemonstrated theses traditionally rooted within Aristotle's conjectures:

> Since I have found the material of springs of the best quality possible, I mean fresh water in plenty, passed through
> and purified by pure and clean sands and that it remains for me only to raise it to the mouths of springs, I no longer
> have any difficulty, all Philosophy, ancient and modern, is with me, and is in agreement that that can be done easily
> and naturally (Perrault, 1967, p.108).

After briefly recalling the main theses on this subject as reported by some philosophers, among which there are Aristotle, Seneca, Lydiat and Descartes, Perrault resumes that:

> . . . all have remained in agreement about this rising of water or of aqueous vapors to the top of mountains; and
> this consensus, although based on different principles, is a fairly good sign of the truth of this fact (Perrault, 1967,
> p.109).

    The fact that Perrault explains the hydrological cycle by coming back to the ancient tradition, enlightens some intricate features of the Scientific Revolution, which did not proceed straight forward, but was characterised by alternative phases of advancement and uncertainty. Nevertheless, with his attitude at moving from the laboratory to the field, his research for shared methods of measurement, his acceptance of quantitative observations until they are not falsified but other researchers,

Perrault interprets the spirit of the Scientific Revolution and seems to anticipate some themes which are peculiar of the modern epistemology of the complexity. We should probably admit that, already in the XVII century, some scientists realised that the nascent paradigm of classical Physics, focused on the perfect repeatability of experiment, was inadequate to deal with some natural phenomena. Therefore, the birth of modern Hydrology implied not only the emergence of the scientific method, but also a supplementary epistemological reflection on the limit of laboratory experiment, thus going beyond the common image

of Scientific Revolution.

    Apart what is reflected in the *Origine*, is it possible to find witnesses of this supplementary epistemological reflection in Perrault's opus? Perrault's Cartesian appeal to simplicity, practice and common sense only apparently clashes with epistemological awareness. In fact, even a cursory look at the *Lettre à Mr. Hu[y]guens au sujet des experiences* (Letter to Mr. Huygens about experiments, written in 1672 and published as appendix of the *Origine* two years later) demonstrates that Perrault deeply

reflected on the limits of scientific method (Perrault, 1967, pp.146–157). In his letter to the great Dutch Physicist, Perrault defends two hypotheses which are wrong. Nonetheless, he presents arguments which are not devoid of interest from the epistemological point of view. Huygens warmly (and correctly) invites him to refuse the old idea of *horror vacui* as explanation for the functioning of pumps and he also suggests him to discard attraction and repulsion as valid physical concepts. Perrault answers with claims which raise some issues for the nascent scientific method, as they call into question the risks of a blind



faith in the experimental results. The core of Perrault's thesis can be briefly sketched as follows: a deep awareness of the limits of experience is essential to a thoroughly scientific approach and the result of a single experiment must never lead the scientists to undue generalizations of their theses. Hereafter, we resume the main arguments he puts forward to support this claim.

First, according to Perrault, the scientist must never declare himself satisfied with the results of an experiment before he has
reached the full agreement between experiences and the judgement:

> For whatever experiments can be performed, one cannot stop there, if judgement and the senses all together do not agree about them: the senses are often mistaken when they act alone, and the judgement is often deceived also sometimes unless the senses set it right (Perrault, 1967, p.147).

Senses alone are necessary but not sufficient in order to obtain effective knowledge, and the same applies for judgement; there-
fore, it is wrong to consider definitive any experiment, insofar as it leaves open the possibility of both different interpretation and further experience. The second argument concerns the concepts of cause and effect: according to Perrault, in Nature no effect is produced by a single cause, rather every effect is the consequence of a multiplicity of causes. He also introduces the distinction between internal causes (which are intrinsic to the dynamic of the phenomena) and external causes (which play an accidental role in the determination of the effect):

> It is certain that in Nature, no effect is produced by a single cause, and that on the contrary there is none without several causes, some of which are particular to the things on which the effects are produced, and the others are alien to them and come from the outside, and nevertheless take part in the production of their effects (Perrault, 1967, p.147).

The third argument calls into question what Perrault calls the "machines", i.e. all the instruments which are used to measure
the phenomena and to assemble the experimental model. As far as he considers the question of measurement absolutely crucial for the success of the experiment, Perrault warns that the discrepancy of proportions between laboratory model and real natural process could compromise the generalisation of experimental results:

> It appears that proportion is absolutely necessary in machines to make them yield the effects desired; and it is likewise obvious that general conclusions cannot be drawn from many experiments that are made, and that all that
may be learned from them, is merely that what they show us, can be done with the machines, the instruments and the materials which we have used; and at the same time to make us fear that in making them with other machines and of other proportions, or with other instruments, other materials and in other circumstances, they might have other results (Perrault, 1967, p.149).

Lastly, Perrault recalls that nobody can exclude further explanations of the old problems, given the progress of both human
mind and technical application, which will probably make available new interpretations and experimental models:

> ...if these experiments have caused us to doubt so many things about which we were, so it seemed, very sure, they should cause us to doubt even more many things which we now believe are very certain, and cause us to fear





that some day posterity will give us the same treatment, and laugh at our Philosophy just as we laugh at that of Antiquity (Perrault, 1967, p.151).

Perrault's thesis about the *horror vacui* has been proved to be wrong, but the questions he rises are noteworthy from many points of view. Above all, in the letter there are neither direct references to his experiments reported in the *Origine*, nor to the

basin hydrology, but the letter was written in 1672, just after the three years in which he was collecting data about the water balance of the Seine river, and probably during the composition of the book. If we consider these circumstances, it might be realistic that the hydrological studies of the *Origine* and the epistemological remarks of the letter deeply influenced each other.

What can be inferred by Perrault's remarks? As for the first argument, it is a classical *topos* of Scientific Revolution, thus it does not depart from the XVII century's paradigm. On the contrary, Perrault's emphasis on the multiplicity of causes may be

connected to his studies to understand the hydrological cycle. It is important to stress that, according to Perrault, the multiplicity of causes does not represent an objection, rather a stimulus for scientific enquiry. Furthermore, there is no contradiction between the epistemology of the letter and the scientific practice of the *Origine*. Indeed in the forethought of the *Origine*, which is dedicated to Huygens, Perrault states that:

What I have said in my letter to Mr. Hu[y]guens, which is at the end of this book, is not to criticize the experiments,

as has been said by a few who had already seen this letter: but only to say that there is no certainty to be drawn from the general conclusion of certain experiments, and on that basis to attribute to certain things certain effects rather than to others, and to exclude from them all other causes, known or unknown (Perrault, 1967, p.11).

As well, Perrault's awareness of the problem of proportions in the design of the laboratory model is remarkable: he is in fact posing a problem that will be faced a pair of centuries later with the theory of hydraulic and mechanical models, based on

dimensional analysis.

More generally, despite his undeniable faith in scientific progress, Perrault is significantly conscious of the risks involved in an acritical and dogmatic acceptance of experimental results. Not only he is aware that scientific results are strictly connected with the opportunities offered by technical progress, also he admits the possibility of new interpretations of the achieved results, thus showing an understanding of science as an indefinitely open and perfectible field of knowledge. Although we can consider

him one of the founders of scientific soil hydrology and one of the progenitors of modern scientific spirit, Perrault did not consider the experiment as an oracle. On the contrary, he clearly saw the bait of underestimating the complexity of phenomena and the influence of the instruments on the object of observation. Far from being a naïve empiricist, he rejects the blind faith in the experimental results as well as the old appeal to authority. To summarise, the *Lettre* is not only a proof of the individual guile and lucidity of the author, it also demonstrates that the birth of quantitative hydrology implied a deep epistemological

reflection on the statute and on the limits of experiment and laboratory model.





# 7    Conclusions

Perrault framed his main opus, the *Origine*, in a deeply novel epistemological approach, which embraced the new experimental and Galilean method, rooted in the quantitative measurements, to integrate the traditional one, based on qualitative observations and on the discussion of conjectures. In the critical section of his opus he presents, in a few pages, a set of four experiments
with which he intended to test two main difficulties he found with the modern understanding of the hydrological cycle, which at that time was about to be accepted. These difficulties are whether the water may spontaneously rise in the soil and form springs, and whether it may easily percolate through the soil until it reaches an impervious layer, where a groundwater table will form and the water level will rise until forming the springs.

The experiments are precisely described and most of the measures are detailed, so that we are able both to repeat them in
the laboratory, and to recognize incomplete data and missing information. Perrault's idea was therefore not only to experience his conjecture, but also to make his experiments repeatable, and in this sense we may regard at them as probably the first modern hydrological experiments. Yet two of his observations misled his interpretation of the hydrological cycle: at first he tried to onset a spring above what would have been the groundwater table, then he did not recognize the importance of the capillary barrier he realised in the experiment, in confining the water percolation and in reducing the consequent recharge of the
groundwater. The repetition of the experiments and the discussion about its interpretation provide interesting cases, useful to let hydrology students train themselves both with the understanding of fundamental aspects of epistemology and of the modern theoretical framework of soil hydrology.

Perrault's misinterpretation of the hydrological cycle also opens up a discussion of some interesting epistemological issues. At first Perrault comes back to accept the undemonstrated theory, rooted in the Aristotelian thought, according to which the
hydrological cycle is mainly the effect of an evaporation and condensation process that takes place in the center of the Earth. This provides some evidence that the Scientific Revolution is not a straightforward line, but, as a great cultural movement, is articulated in alternative phases of advancement and uncertainty. Then, the problems posed by soil hydrology, even at an early stage, appear irreducible to the classical paradigm of the Scientific Revolution. Even if the experiment designed by Perrault has proved to be repeatable, an in–depth analysis of them shows that the questions posed on the hydrological cycle and on the
soil hydrology are hardly reproducible by means of a controlled laboratory model.

Perrault's scientific practice, which was many–sided, confronted him with complex issues, as the hydrological cycle and the catchment are. This objective complexity seems to be reflected in his letter to Huyguens which, although often disregarded, contains remarkable ideas anticipating a modern epistemology. Perrault should therefore be considered not only an amateur scientist, but also a refined scholar who worked at a turning point within the developing epistemology of the Scientific Revolu-
tion, being aware of its uncertainties and conflicts. In one of his essays, the novelist and literary critic Italo Calvino gave several definitions of what in his opinion is a classic. Among these, the sixth one is: "A classic is a book which has never exhausted all it has to say to its readers" (Calvino, 1991, p.5). As a conclusion we may state that Perrault's *Origine* attractively meets this definition, as so many years after its first appearance still deserves intriguing stimuli and suggestions.





*Data availability.* Data of the experiment will be fully reported in the technical report Berta and Barontini (in preparation) available at the Università degli Studi di Brescia, Dipartimento di Ingegneria Civile, Architettura, Territorio, Ambiente e di Matematica

## Appendix A: Relationships used to characterize soil hydrological properties

### A1 Hydraulic conductivity at soil saturation

The hydraulic conductivity $K_s$ at soil saturation was estimated by means of Kozeny and Carman's relationship:

$$K_s = \frac{\gamma}{\mu} \frac{\tilde{d}^2}{180} \frac{\phi^3}{1 - \phi^2} \,, \tag{A1}$$

in which $\gamma \, [MLT^{-2}L^{-3}]$ is the water unitary weight, $\mu \, [ML^{-1}T^{-1}]$ is the dynamic water viscosity, and the term:

$$k_p = \frac{\tilde{d}^2}{180} \frac{\phi^3}{1 - \phi^2} \tag{A2}$$

is the intrinsic permeability $k_p \, [L^2]$. In Eq. (A2) $\tilde{d} \, [L]$ is a characteristic grain diameter of the soil and $\phi \, [L^3 L^{-3}]$ is the soil

porosity. Knowing the grain size distribution curve of the soil, the characteristic diameter $\tilde{d}$ was estimated by means of the weighted harmonic average of the grain size distribution curve diameters $d_i$:

$$\tilde{d} = \frac{1}{\sum_i (W_{i+1} - W_i) \frac{1}{2} \left( \frac{1}{d_{i+1}} + \frac{1}{d_i} \right)} \,, \tag{A3}$$

where the weights $W_i \, [MM^{-1}]$ are the fraction of the soil passing throw the sieve with diameter $d_i$. Therefore $W_{i+1} - W_i$ is the soil fraction retained by the sieve $d_i < d_{i+1}$ and $\sum_i (W_{i+1} - W_i) = 1$.

### A2 Soil–water retention relationships

Soil–water retention relationships were represented in the forms proposed by Brooks and Corey (1964) and by van Genuchten (1980). Brooks and Corey's relationship is:

$$s = \begin{cases} \left( \frac{\Psi}{\Psi_b} \right)^{-\lambda} & \Psi \leq \Psi_b \\ 1 & \Psi_b < \Psi \leq 0 \end{cases} \,, \tag{A4}$$

where $s \, [1]$ is the effective soil saturation, $\Psi \, [MLT^{-2}L^{-2}]$ is the soil–water pressure, which may expressed also in potential

form i.e. with $[L]$–dimension, $\Psi_b$ is the bubbling pressure, i.e. the minimum value of the pressure potential for which the gaseous phase is discontinuous, and $\lambda$ is the pore–size distribution index. Generally the greater $\lambda$, the greater is the soil–grain uniformity.

Van Genuchten's relationship is:

$$s = \frac{1}{\left[ 1 + \left( \frac{\Psi}{\Psi_1} \right)^n \right]^m} \tag{A5}$$





where $\Psi_1$ is a scaling factor and $n$, $m$ are shape coefficients. According to van Genuchten (1980), in order to analytically integrate Mualem's predictive form for the conductivity of the unsaturated soil, the shape coefficients are constrained by the relationship:

$$m = 1 - \frac{1}{n}. \tag{A6}$$

5  If $m$ and $n$ are not constrained, the soil–water retention curve (A5) has an inflexion point $\Psi_f$ at:

$$\frac{\Psi_f}{\Psi_1} = \sqrt[n]{\frac{n-1}{mn+1}}. \tag{A7}$$

In case Eq. (A6) holds, the relationship (A7) takes the form:

$$\frac{\Psi_f}{\Psi_1} = \sqrt[n]{\frac{n-1}{n}} \equiv m^{\frac{1}{n}}. \tag{A8}$$

*Competing interests.*  The authors declare no competing interests regarding the submission of this article.



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

Vitruvius and Perrault, C.: Les dix livres d'architecture, Jean–Baptiste Coignard, Paris, 1673.



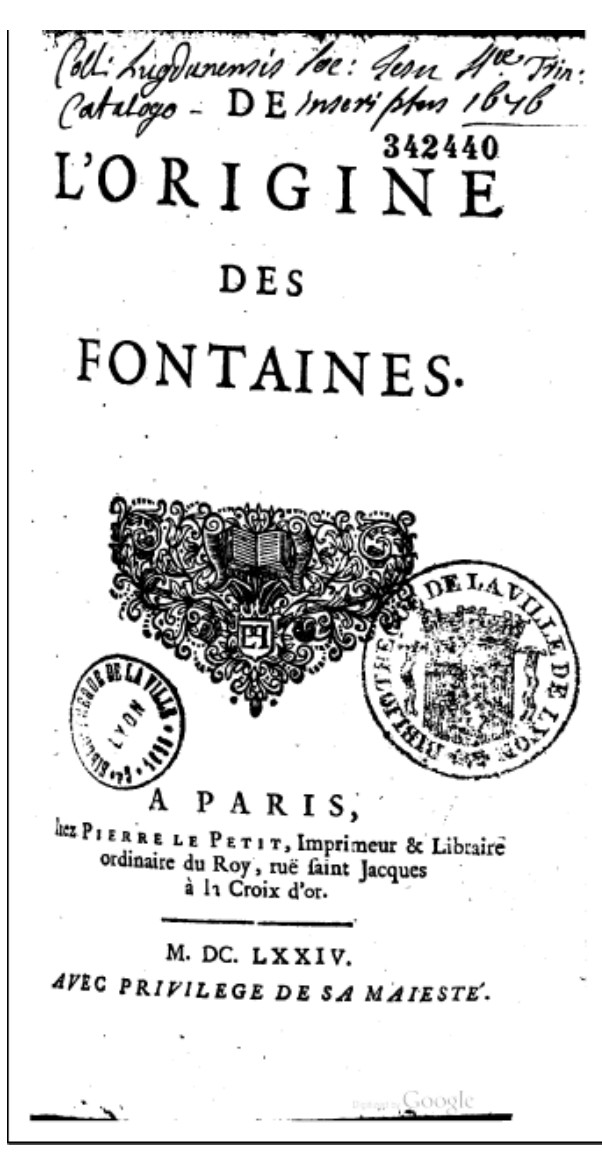

**Figure 1.** Front matter of Perrault's opus *De l'origine des fontaines* (Perrault, 1674, source: https://books.google.com, Library Project, accessed 12/8/2019)





**Figure 2.** Front matter of Schott's opus *Anatomia physico–hydrostatica fontium ac fluminum* (Schott, 1663, source: https://books.google.com, Library Project, accessed 12/8/2019)





**Figure 3.** Statement of Perrault's opinion (Perrault, 1674, p.148, source: https://books.google.com, Library Project, accessed 12/8/2019)



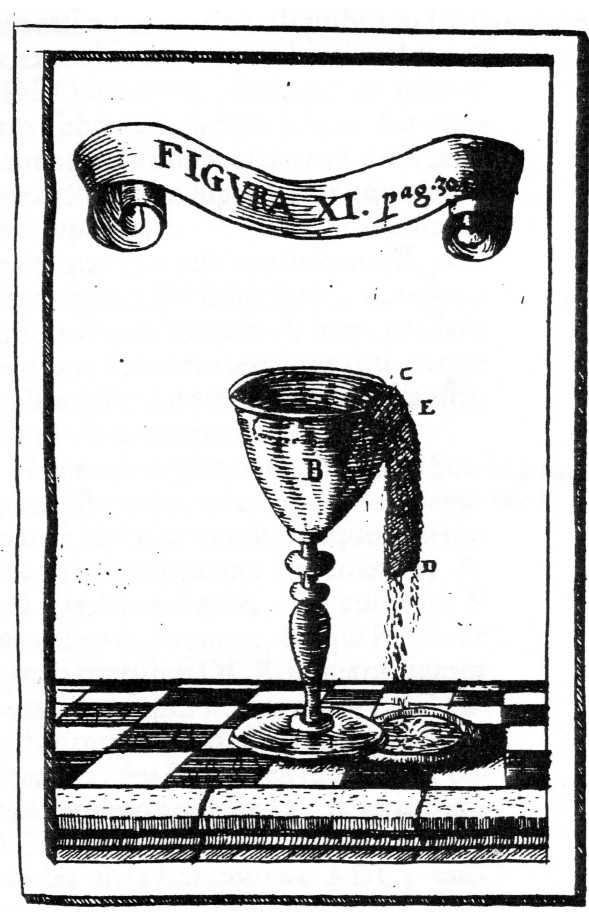

**Figure 4.** Figure XI from Schott's opus *Anatomia physico–hydrostatica fontium ac fluminum* (Schott, 1663, source: https://books.google.com, Library Project, accessed 12/8/2019)




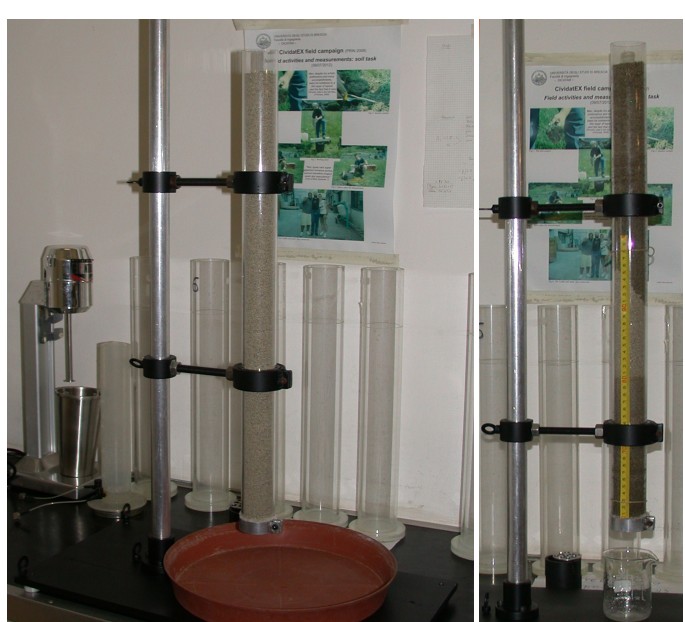

**Figure 5.** Laboratory setup of the soil column (left) and the soil column during the second part of the first experiment, when the top and the bottom imbibition fronts meet (right)





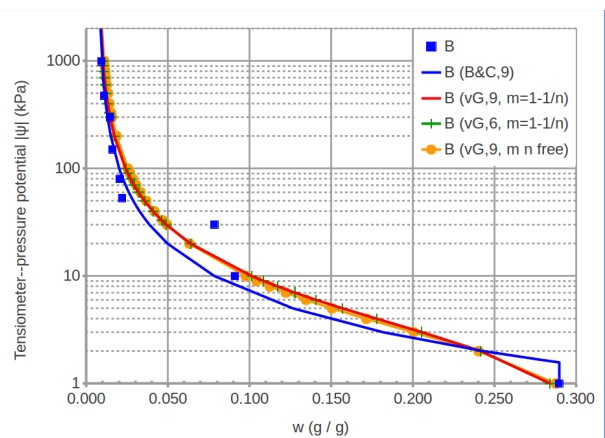

**Figure 6.** Experimental and interpolated soil–water retention relationships of one of the sandy soils used to repeat Perrault's experiments (B&C: Brooks and Corey's curve, vG: van Genuchten's curve; 6 and 9 are the number of experimental points used to interpolate the analytical curves)



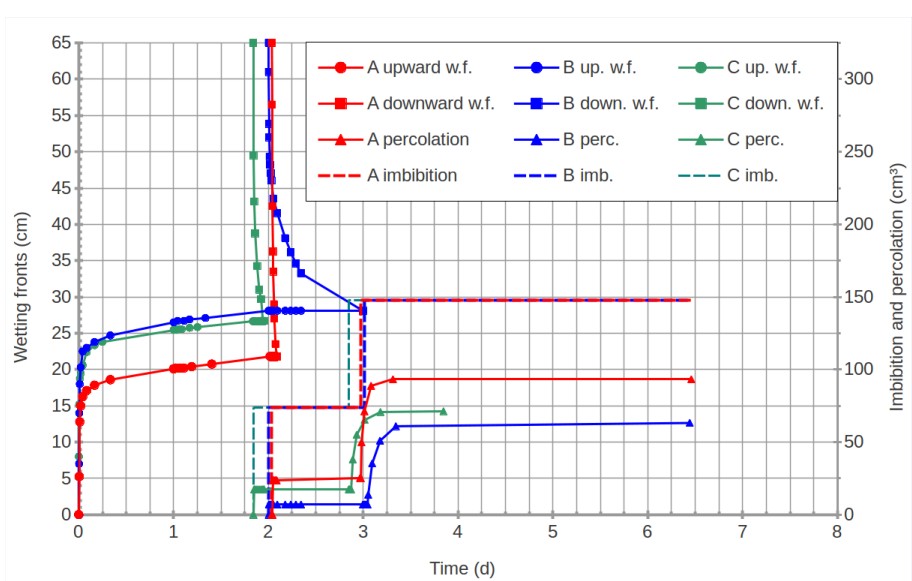

**Figure 7.** Measures of the position of the water fronts, of imbibition and of percolation collected during the three repetitions of the first experiment



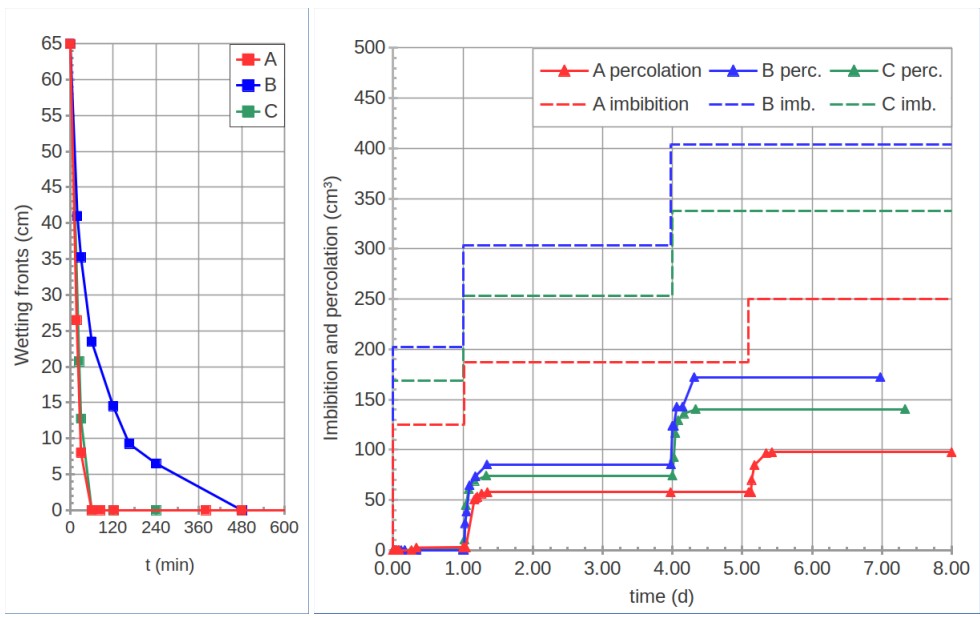

**Figure 8.** Measures collected during the three repetitions of the third experiment: depth of the imbibition fronts (left) and volumes of imbibition and percolation (right)



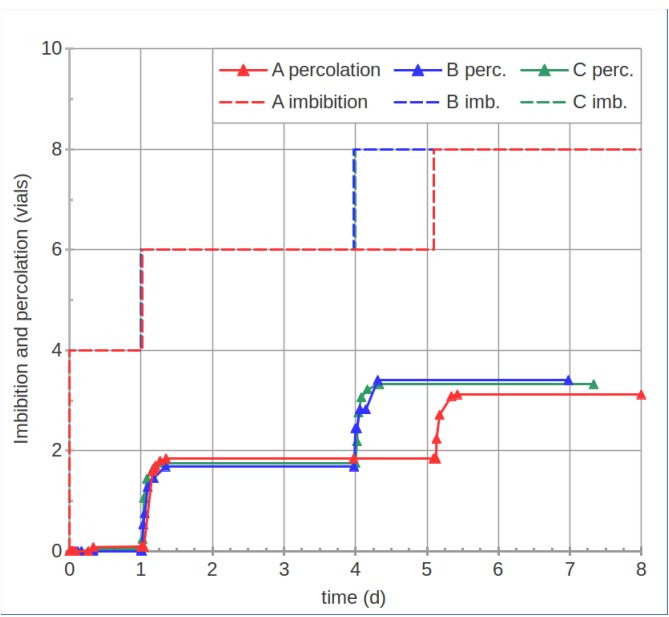

**Figure 9.** Imbibition and percolation volumes during the three repetitions of the third experiment. The volumes are divided for the unitary volume $V_i$ estimated for each soil



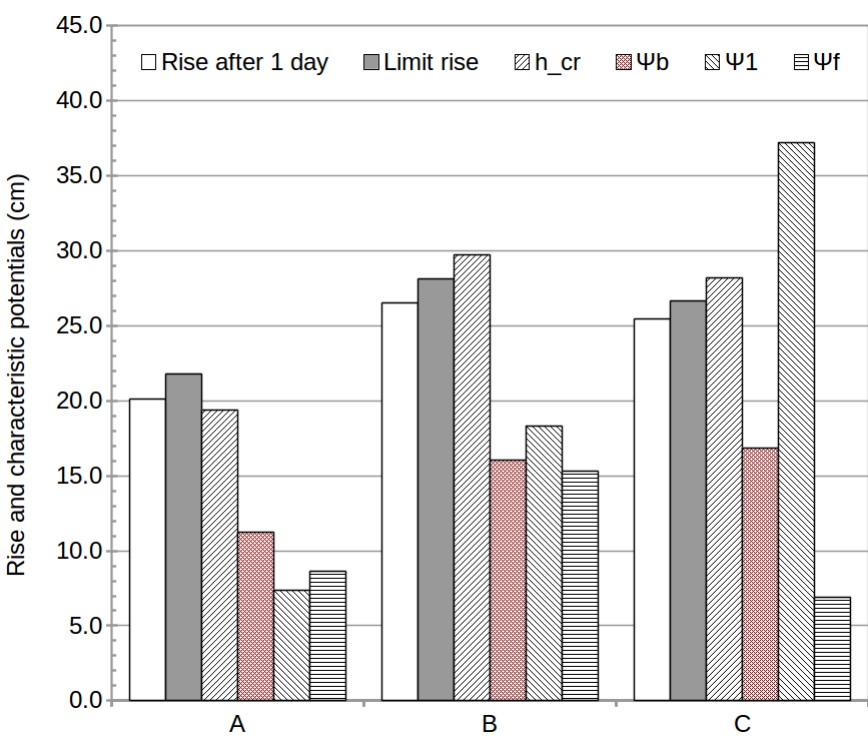

**Figure 10.** Comparison between the measured capillary rise after 1 day, the limit capillary rise, the capillary rise $h_{cr}$ estimated by means of Polubarinova–Kochina formula and the absolute values of the characteristic potentials $\Psi_b$, $\Psi_1$, $\Psi_f$ for the three soils used to repeat Perrault's experiments



**Table 1.** List of previous scholars referred to by Schott (1663) and by Perrault (1674)

| Scholar | Schott (1663) | Perrault (1674) |
| --- | --- | --- |
| Agricola | p.110 | p.48 |
| Albertus Magnus | pp.117–121 | |
| Aristotle | pp.66–76 | pp.13–21 |
| Cabeus Nicolas | pp.111–114 | |
| Cardano | | pp.40–49 |
| Coimbran philosophers | pp.106–109 | pp.35–38 (together with Thomas Aquinas) |
| Cornelius a Lapide | pp.114–117 | |
| Davity | | pp.89–90 |
| Descartes | pp.82–84 | pp.90–97 |
| Dobrzenzki de Nigro Ponte | | pp.49–51 |
| Du Hamel | | pp.117–124 |
| Epicurus | | pp.21–22 |
| Froidmond Libert | pp.109–110 | |
| Gassendi | | pp.116–117 [‡] |
| Holy Scriptures and Patristics | pp.88–102 | |
| p. (père) Jean François | | pp.142–145[‡] |
| p. (père) Lennart Leys (Leonardus Lessius) | pp.77–78 | |
| Lucretius | pp.96–99 | |
| Maignan | pp.126–128 | (see Schott) |
| Palissy | | pp.145–146 [‡] |
| Papin | | pp.97–116 |
| Presocratic philosophers | p.74 | |
| Plato | pp.49–53, 78 | p.8 |
| Plinius | | pp.32–35 |
| Rohault | | pp.138–142 |
| Scaliger (Julius Caesar Scaliger) | | pp.38–40 |
| Schott | | pp.115, 124–138 |
| Scotus Duns | pp.121–125 | |
| Seneca | pp.77, 86–88 | pp.24–32 |
| Other scholars reported by Seneca | p.85 | pp.27–30 |
| Thomas Lydiat | pp.129–130 | pp.72–88 |
| Vallès Francisco (Vallesius) | pp.103–106 | pp.33–34 |
| Vitruve | | pp.22–24 [‡] |
| von Helmont | | pp.51–72 |

[‡] Authors referred to as *Opinion Commune* by Perrault





**Table 2.** Measurement systems used in the XVII–century France (Blamont, 2001)

| Measurement unit | *pied du roi* | *Toise de l'Ècritoire* (before 1667) | *Toise du Châtelet* (since 1668) |
|---|---|---|---|
| *point* | 1/1728 | 0.189 mm | 0.188 mm |
| *ligne* | 1/144 | 2.268 mm | 2.256 mm |
| *pouce* | 1/12 | 2.722 cm | 2.707 cm |
| *pied du roi (pied)* | 1 | 32.660 cm | 32.484 cm |
| *toise* | 6 | 1.959 m | 1.949 m |

**Table 3.** Perrault's description of the soils used during the first and second experiment

| | Perrault's description |
|---|---|
| 1st | Dry river sand, passed with coarse sieve |
| 2nd | Sand with gravel |
| 3rd | Crumbled and sieved sandstone |
| 4th | Crumbled granular soil |

**Table 4.** Capillary rise and characterization of Perrault's soils ($1^{st}$, $2^{nd}$, $3^{rd}$, $4^{th}$) and of the soils used for the repetitions of the first experiment (A, B, C). Legend: $\phi$ porosity, $d_{10}$ characteristic diameter for Polubarinova–Kochina's relationship, $\tilde{d}$ characteristic diameter for Kozeny and Carman's relationship, $h_{cr}$ Polubarinova–Kochina's capillary rise, $k_p$ Kozeny and Carman's intrinsic permeability, $K_s$ Kozeny and Carman's hydraulic conductivity at saturation. Perrault's soils are describe in Table 3

| | $\phi$ | $d_{10}$ | $\tilde{d}$ | Perrault's data | After one day | Limit rise | $h_{cr}$ | $k_p$ | $K_s$ |
|---|---|---|---|---|---|---|---|---|---|
| | cm$^3$/cm$^3$ | cm | cm | pouces | cm | cm | cm | cm$^2$ | m/s |
| 1st | 0.333 | 0.0185 | (0.0185) | 18 | 48.7 | | | 1.58E-07 | 1.55E-04 |
| 2nd | 0.333 | 0.0332 | (0.0332) | 10 | 27.1 | | | 5.11E-07 | 5.01E-04 |
| 3rd | 0.333 | 0.0332 | (0.0332) | 10 | 27.1 | | | 5.11E-07 | 5.01E-04 |
| 4th | 0.333 | 0.0185 | (0.0185) | 18 | 48.7 | | | 1.58E-07 | 1.55E-04 |
| A | 0.463 | 0.0268 | 0.0444 | | 20.1 | 21.8 | 19.4 | 3.77E-06 | 3.70E-03 |
| B | 0.478 | 0.0165 | 0.0241 | | 26.5 | 28.1 | 29.7 | 1.29E-06 | 1.27E-03 |
| C | 0.440 | 0.0198 | 0.0340 | | 25.45 | 26.65 | 28.2 | 1.74E-06 | 1.71E-03 |





**Table 5.** Imbibition and percolation volumes measured during the first experiment. Data are reported as number of vials, in order to make a comparison with Perrault's data. The unitary volume of a vial was estimated equal to $74\,cm^3$ for all the soils, according to Barontini et al. (2013)

| Imbibition (vial) | Perrault's data (vial) | A (vial) | B (vial) | C (vial) |
|---|---|---|---|---|
| 1 | 0.75 | 0.34 | 0.09 | 0.24 |
| 1 | 1 | 0.92 | 0.76 | 0.73 |

**Table 6.** Imbibition and percolation volumes measured during the third experiment. Data are reported as number of vials, in order to make a comparison with Perrault's data. The unitary volume $V_i$ of a vial was estimated equal to $V_i = 31.3, 50.5, 41.0\,cm^3$ for the soil A, B, and C, respectively, according to the procedure described in Section 4

| Time (d) | Imbibition (vial) | Perrault's data (vial) | A (vial) | B (vial) | C (vial) |
|---|---|---|---|---|---|
| 0 | 4 | 0.33 | 0.09 | 0 | 0.04 |
| 1 | 2 | 2 | 1.76 | 1.69 | 1.72 |
| 4 | 2 | 1.75 | 1.36 | 1.72 | 1.61 |