# Peer review of "Beyond Perrault's experiments: Repeatability, didactics and complexity"

_Hydrology and Earth System Sciences, 2019_

## Referee Comment (RC1) · Dani Or (Referee) · 26 Sep 2019

1. Generally, the treatment tends to be superficial and not as definitive as the account of Raymond Nace 1974 (Wat. Res. Bull. Am. Wat. Res. Ass.) regarding Pierre Perrault's contributions to hydrology (and, as the authors argue, to systematic experimental approaches to hydrology).

2. The attribution of the experiments that formed the basis of his early research, namely the capillary rise observations of Magnanus (Maignan) as reported by Schott are very sketchy, and despite their importance are poorly attributed

3. Considering the focus on epistemology advanced by the authors, I find it remarkable that the authors have chosen to make focus on Aristotle in reference to to ancient school of thoughts when a pivotal book by Lucretius Caro (De Rerum Nature - on the nature of things) discusses much more definitively the basics and far more advanced concepts relevant to hydrology than Aristotle's work (Caro was influenced by the scientific school of Epicurus and wrote about 200 years after Aristotle published "Meteorology" ). For example, the role of the sun in the process of evaporation was well known in Lucretius Caro's time (99-55 BC) as he writes "The same, spread out before the sun, will dry; Yet no one saw how sank the moisture in, Nor how by heat off-driven. Thus we know, That moisture is dispersed about in bits Too small for eyes to see"

4. We all agree that the work of Perrault was very important for modern hydrology as a scientific discipline – hence, it seems a bit odd that the authors have chosen to "recycle" an old exercise by the main authors to focus on unsaturated flow parameterization towards explaining the incomplete understanding of unsaturated flow at the time (it seems similar to Philip et al (1989, WRR - https://doi.org/10.1029/WR025i001p00016 that relished the "surprise" of Perrault for not finding water in the buried clay pot experiment. . .. ) In other words, both the Hydrus experiment of Barontini, and the general tendency to glorify present state of knowlöedge regarding the unsaturated state does not add much insight nor appreciation of the the historically significant work of Pierre Perrault. In my view, the parameterization and interpretation (through modern tools) of unsaturated flow are detrimental to the main story of this paper

6. I am not convinced by the argument that Perrault pioneered and contributed to complexity, perhaps to experimentation – a more critical (and much earlier) thinker was Lucretius Caro that anticipated modern hydrology (and atomistic theories) – but this is my view. . .

7. Since this is an opinion piece with subjective interpretation of the work and Pierre Perrault to early hydrology and thus is a historical narrative – I am not in a position to ask the authors to write their piece the way I would have liked to see, but I hope the

comments are useful.

---

## Author Comment (AC1) · 7 Oct 2019

We thank Prof. Dani Or for the comments. We will account for them in a revised version of the paper, as we report in the point–to–point reply:

1. Nace's article of 1974 (doi:10.1111/j.1752-1688.1974.tb05623.x) represented an important source for our paper, however the aim and methods of the two papers are essentially different. Nace presented both a thorough historical and biographical account of Perrault's life and a detailed summary of the content of Perrault's book. Our focus is rather on the epistemological value of Perrault's experiment and on the relationship between his work and Scientific Revolution. For exam-

ple we analysed Perrault's text and found relevant correspondences between his sentences and that of thinkers such as Pascal and Descartes. Moreover we analysed Perrault's view on experiments and experimental method as expressed in the letter to Mr. Huygens, which has been quite ignored by previous works on the argument.

2. Maignan's experiment is reported in the third part (*Philosophia naturalis*, *Natural philosophy*, Chapter 14, Proposition 14) of his *Cursus philosphicus...* (*Philosophical course...*) published in 1653. In a revised version of the paper we will add the full reference to Maignan's experiment and a little more details, but his report is not much more detailed than what is reported by Schott (1663), most of which was already reported in our paper. Moreover it is noteworthy that Maignan presents his experience in the context of a wider discussion *De gravi & levi*, i.e. *On the gravity of things*, deeply framed in Aristotelian Physics, and his experience is not directly referred to a discussion on the hydrological cycle. It was Schott who took Maignan's experience and used it to support his dissertation on the hydrological cycle.

3. We agree that Lucretius had a quite advanced understanding of the water cycle, but we did not focus on his work, because he is not mentioned in Perrault's book. In fact Perrault cursory mentioned Epicurus, while a reference to Lucretius may be found in Schott. However they did not enter in detail of the comparison between their own understanding of the hydrological cycle and Epicurus' and Lucretius' ones, as their main term of reference is always the Aristotelian tradition, which was followed by many scholars until Perrault's age and beyond.

4. In our opinion, relevant keys to enlighten the cultural importance of Perrault's experiments, also by an epistemological point of view, are their repeatability and their didactic implications for modern students. Moreover, far from the idea both of glorifying modern models of soil hydrology with respect to ancient understandings, and of glorifying Perrault's experiments with modern frameworks, we think that training ancient conclusions with modern theories is a constructive exercise to deeply understand ancient Authors' comprehension of the phenomena.

This is why we presented (for the first time, the laboratory technical report being on its way to be published) some results of the laboratory repetition of the experiments. The fact that we may qualitatively and, at some extents, quantitatively repeat Perrault's experiments, more than three centuries later, emphasizes their relevance in the context of the Scientific Revolution.

Then, we proposed a discussion on the didactic implications of teaching the soil hydrology through Perrault's experiments, and in this context we reported some aspects of a numerical experiment we performed some years ago. Yet we did not reported the results of the reanalysis, as they have already been published in the referred work (Barontini et al., 2013, doi:10.1016/j.proenv.2013.06.067). Here our focus was to propose a discussion on the framework that was followed, with respect to its didactic usefulness.

In the present version of the paper we did not refer to the buried clay pot experiment, because according to Nace (1974) it was not performed by Pierre Perrault, but probably by Pierre's brother Charles and by Philippe de la Hire (Philip et al., 1989, doi:10.1029/WR025i001p00016). Yet it may be interesting to contextualize Pierre's importance in the context of his age, and we will refer to it in a revised version of the paper.

5. We agree that it would be excessive to define Perrault as an epistemologist of complexity: it is always important, while discussing such issues, to bear in mind the historical context. However, Perrault's experience in soil hydrology led him to general reflections on the limits of experimental method. These reflections are not our subjective opinions: in the letter to Mr. Huygens Perrault clearly and openly expresses his perplexities about the blind faith in experiments that seems to dominate some of his contemporaries. As far as we know, nobody has yet

focused on this aspect of Perrault thought, and we find it interesting because it partially clashes with the typical image of a pioneer in Scientific Revolution.

---

## Referee Comment (RC2) · Okke Batelaan (Referee) · 13 Nov 2019

General comment The manuscript 'Beyond Perrault's experiments: Repeatability, didactics and complexity' by Stefano Barontini and Matteo Settura is submitted for the special issue of HESS on 'History of Hydrology'. It is an interesting and appropriate submission for this journal and special issue. In the manuscript the authors try to raise attention to aspects of experiments performed by Perrault, which are not well known so far, hence the 'Beyond' in the title. These aspects: repeatability of experiments, didactical value of the experiments and complexity are of general interest for the history of science and in particular for the development of hydrology as a field of science. So, the

idea of the paper is good. However, the paper suffers from a number of less than optimal presentation aspects due to which its current accessibility (readability) and impact suffers. The main problems can be summarized as: 1: The language and structure of the text is not good. It would strongly benefit from a detailed English editing and some rewriting of sections to make certain aspects clearer. 2: Although the paper does not focus on the main scientific achievements of Perrault, the authors cannot assume that the reader is fully aware of these. They may have been highlighted in the paper of Nace (1974), however that paper might not be directly available (I had to order it). Currently, this manuscript provides in a sketchy way the main achievements of Perrault to the reader. I suggest that upfront in the introduction a better summary of Perrault's hydrological science achievements are presented. Next the paper can continue in more detail on the less well known aspects of some of these achievements. Specifc comments: Does the paper address relevant scientific questions within the scope of HESS? YES Does the paper present novel concepts, ideas, tools, or data? YES Are substantial conclusions reached? YES Are the scientific methods and assumptions valid and clearly outlined? YES Are the results sufficient to support the interpretations and conclusions? YES Is the description of experiments and calculations sufficiently complete and precise to allow their reproduction by fellow scientists (traceability of results)? YES Do the authors give proper credit to related work and clearly indicate their own new/original contribution? I suggest that the authors add the reference to English translation of Perrault's work. This makes it possible for interested readers to trace and read the work itself. La Roque, A. (1967). On the origin of springs, Pierre Perrault (translation). New York and London, Hafner Publishin.g Co. Does the title clearly reflect the contents of the paper? YES Does the abstract provide a concise and complete summary? NO: the abstract does not represent a proper summary of the main text, currently it reads more like an introduction. It needs to be completely rewritten. Is the overall presentation well structured and clear? NO: The introduction should better explain/summarize the main achievements of Perrault as known and described by e.g. Nace (1974). Then is should explain what the issue is that is not known well about Perrault and how this paper will enlighten us in showing these aspects of Perrault's work. Is the language fluent and precise? NO: The English grammar and spelling needs to be improved. Now there are sentences, which are difficult to understand or where the formulation is imprecise. E.g. the very first words of the abstract state: 'The studies conducted. . .', which studies do the author refer to? The style of writing is also with a lot of long sentences and paragraphs, which makes it more difficult to follow. An example is p6, line 23-27. Line 4 page 2, '. . .in order to become a modern science, Hydrology needed two steps. . .'; I do not like this style of de-personified subjects. Sciences, including hydrology is advanced by people, crediting then the science itself with steps 'needed' is, especially in a historical oriented paper, to be avoided. Line 21 page 6, experiences? Or experiments?

Are mathematical formulae, symbols, abbreviations, and units correctly defined and used? NO: not all symbols of the equations are explained. Should any parts of the paper (text, formulae, figures, tables) be clarified, reduced, combined, or eliminated? YES: Abstract and introduction should be rewritten. I did not find the explanations on the experiments (section 4 and 5) and the accompanying figures very clear, I suggest some strong editing of these sections as well. Are the number and quality of references appropriate? Add reference to La Roque Is the amount and quality of supplementary material appropriate? FINE

---

## Author Comment (AC2) · 6 Dec 2019

We thank Prof. Okke Batelaan for the appreciation and the suggestions. We will account for the suggestions in a revised version of the paper, as we report in the point–to–point reply. For the sake of clarity, recalls of Reviewer's suggestions are *emphasized* in the following notes:

1. *The language and structure of the text. . .* and *Although the paper does not focus. . .*: We will review the paper according to the suggestions, please see below for some details;

[Figure]

2. *. . . add the reference to the English translation of Perrault's work*: All the citations of Perrault's opus have already been taken from La Roque's 1967 translation, but we chose to cite it as Perrault (1967) and we did not note that the compiler missed the translation note in the references section. . . In a revised version we will explicitly recall the translation so that it will be findable by the reader;

3. *The abstract does not represent. . .*: The abstract will be modified according to the suggestion. Particularly we will shorten the first part (first two paragraphs), relieving its introductory style and better focusing on Perrault's opus. Than we will better evidence which are the focus and the original contribution of the paper in the second part.

4. *The Introduction should better explain/summarize*: We will reorganize the Introduction according to the suggestion of firstly recalling Perrault's main achievements and enlightening Nace's (1974) contribution to their comprehension, and then of better detailing our focus. This will develop the first current 20 lines, which will increase in length. Therefore we plan also to enucleate the analysis of the cultural context at Perrault's ages in a standing–alone section between the Introduction and current Section 2;

5. *The English grammar and spelling needs. . .*: English grammar and spelling will be thoroughly reviewed, too long sentences will be cut in order to make the paper better readable. In the current version of the paper we capitalized the subjects (Hydrology, Physics, etc.) according to a common usage in the historical and philosophical milieu. We would prefer to maintain this usage in a revised manuscript, but we will pay attention when it is misleading, as in the cases raised by the Reviewer, and in that cases we will rephrase the sentence;

6. *The studies conducted. . .*: We referred to naturalistic and philosophical studies. The sentence will be rephrased while reviewing the abstract;

7. *P.6, line 21: Experiences?*: The word will be substituted with *experiments*, thanks for the note;

8. *Not all symbols...*: We will check all the symbols. Many of them were described in the Appendix, where most of the the equations are presented, but in doing so many symbols may be recognized after that they are found within the text. In a revised version of the paper we will fix this issue.

9. *... explanations of the experiments not very clear*: Sections 4 and 5 (actual numbers) will be reviewed in order to avoid such a lack of clarity;

10. *Add reference to La Roque*: Please refer to item 2 in this list.

---

## Author Response (AR1)

**Reply to Reviewer's comments**

Stefano Barontini     Matteo Settura

January 17, 2020

**1  Reply to Reviewer #1 (Dani Or)**

We thank Prof. Dani Or for the comments. We accounted for them in the revised version of the paper, as we report in the point–to–point reply. For the sake of clarity, recalls of Reviewer's suggestions are *emphasized* in the following notes. The reply includes Authors' Comment #1 previously published in the discussion, as a starting point.

1. *Generally, the treatment tends to be superficial and not as definitive as the account of Raymond Nace 1974 (Wat. Res. Bull. Am. Wat. Res. Ass.) regarding Pierre Perrault's contributions to hydrology (and, as the authors argue, to systematic experimental approaches to hydrology).*

   Nace's article of 1974 (doi:10.1111/j.1752-1688.1974.tb05623.x) represented an important source for our paper, however the aim and methods of the two papers are essentially different. Nace presented both a thorough historical and biographical account of Perrault's life and a detailed summary of the content of Perrault's book. Our focus is rather on the epistemological value of Perrault's experiment and on the relationship between his work and Scientific Revolution. For example we analysed Perrault's text and found relevant correspondences between his sentences and that of thinkers such as Pascal and Descartes. Moreover we analysed Perrault's view on experiments and experimental method as expressed in the letter to Mr. Huygens, which has been quite ignored by previous works on the argument. In the present version of the paper, we better recalled Perrault's main achievements in the Introduction, on the basis of Nace's paper, as it was suggested also by Reviewer #2.

2. *The attribution of the experiments that formed the basis of his early research, namely the capillary rise observations of Magnanus (Maignan) as reported by Schott are very sketchy, and despite their importance are poorly attributed*

   Maignan's experiment is reported in the third part (*Philosophia naturalis*, *Natural philosophy*, Chapter 14, Proposition 14) of his *Cursus philosphicus...* (*Philosophical course...*) published in 1653. In the present version of the paper we added the full reference to Maignan's experiment and some more details. Yet his report is as detailed as what is reported by Schott (1663), most of which was already reported in the previous version of the paper. Both Maignan and Schott complete their comment with a brief discussion on how a fluid in a sponge behaves as the water in the soil, but it it is noteworthy that Maignan presents his experience in the context of a wider discussion *De gravi & levi*, i.e. *On the gravity of things*, deeply framed in Aristotelian Physics, and his experience is not directly referred to a discussion on the hydrological cycle. It was Schott who took Maignan's experience and used it to support his dissertation on the hydrological cycle. We preferred to omit the description of the rise in a sponge, because in our opinion did not added more regarding Maignan's and Schott's comprehension of water hydrology, than it was already expressed by the citation.

3. *Considering the focus on epistemology advanced by the authors, I find it remarkable that the authors have chosen to make focus on Aristotle in reference to to ancient school of thoughts when a pivotal book by Lucretius Caro (De Rerum Nature - on the nature of things) discusses much more definitively the basics and far more advanced concepts relevant to hydrology than Aristotle's work (Caro was influenced by the scientific school of Epicurus and wrote about 200 years after Aristotle published Meteorology ). For example, the role of the sun in the process of evaporation was well known in Lucretius Caro's time (99-55 BC) as he writes «The same, spread out before the sun, will dry; Yet no one saw how sank the moisture in, Nor how by heat off-driven. Thus we know, That moisture is dispersed about in bits Too small for eyes to see.»*

We agree that Lucretius had a quite advanced understanding of the water cycle, but we did not focus on his work, because he is not mentioned in Perrault's book. In fact Perrault cursory mentioned Epicurus, while a reference to Lucretius may be found in Schott. However they did not enter in detail of the comparison between their own understanding of the hydrological cycle and Epicurus' and Lucretius' ones, as their main term of reference is always the Aristotelian tradition, which was followed by many scholars until Perrault's age and beyond.

4. *We all agree that the work of Perrault was very important for modern hydrology as a scientific discipline – hence, it seems a bit odd that the authors have chosen to "recycle" an old exercise by the main authors to focus on unsaturated flow parameterization towards explaining the incomplete understanding of unsaturated flow at the time (it seems similar to Philip et al (1989, WRR - https://doi.org/10.1029/WR025i001p00016 that relished the "surprise" of Perrault for not finding water in the buried clay pot experiment. . . ) In other words, both the Hydrus experiment of Barontini, and the general tendency to glorify present state of knowledge regarding the unsaturated state does not add much insight nor appreciation of the the historically significant work of Pierre Perrault. In my view, the parameterization and interpretation (through modern tools) of unsaturated flow are detrimental to the main story of this paper*

In our opinion, relevant keys to enlighten the cultural importance of Perrault's experiments, also by an epistemological point of view, are their repeatability and their didactic implications for modern students. Moreover, far from the idea both of glorifying modern models of soil hydrology with respect to ancient understandings, and of glorifying Perrault's experiments with modern frameworks, we think that put ancient conclusions at a test with modern theories is a constructive exercise to deeply understand ancient Authors' comprehension of the phenomena.

This is why we presented (for the first time, the laboratory technical report being on its way to be published) some results of the laboratory repetition of the experiments. The fact that we may qualitatively and, at some extents, quantitatively repeat Perrault's experiments, more than three centuries later, emphasizes their relevance in the context of the Scientific Revolution.

Then, we proposed a discussion on the didactic implications of teaching the soil hydrology through Perrault's experiments, and in this context we reported some aspects of a numerical experiment we performed some years ago. Yet we did not reported the results of the reanalysis, as they have already been published in the referred work (Barontini et al., 2013, doi:10.1016/j.proenv.2013.06.067). Here our focus was to propose a discussion on the framework that was followed, with respect to its didactic usefulness.

In the revised version of the paper added a reference to the buried clay pot experiment, performed by Pierre's brother Charles and by Philippe de la Hire (Philip et al., 1989, doi:10.1029/WR025i001p00016), in view of contextualizing Pierre's importance in the context of his age. At the same point we referred also to Mariotte's recognition of the importance of macropores for percolation.

5. *I am not convinced by the argument that Perrault pioneered and contributed to complexity, perhaps to experimentation – a more critical (and much earlier) thinker was Lucretius Caro that anticipated modern hydrology (and atomistic theories) – but this is my view. . .*

We agree that it would be excessive to define Perrault as an epistemologist of complexity: it is always important, while discussing such issues, to bear in mind the historical context. However, Perrault's experience in soil hydrology led him to general reflections on the limits of experimental method. These reflections are not our subjective opinions: in the letter to Mr. Huygens Perrault clearly and openly expresses his perplexities about the blind faith in experiments that seems to dominate some of his contemporaries. As far as we know, nobody has yet focused on this aspect of Perrault thought, and we find it interesting because it partially clashes with the typical image of a pioneer in Scientific Revolution.

6. *Since this is an opinion piece with subjective interpretation of the work and Pierre Perrault to early hydrology and thus is a historical narrative – I am not in a position to ask the authors to write their piece the way I would have liked to see, but I hope the comments are useful.*

We hope that this new version of the paper contributes to clarify our perspective on Perrault's opus.

**2 Reply to Reviewer #2 (Okke Batelaan)**

We thank Prof. Okke Batelaan for the appreciation and the suggestions. We accounted for them in the revised version of the paper, as we report in the point–to–point reply. For the sake of clarity, recalls of Reviewer's suggestions are *emphasized* in the following notes. The reply includes Authors' Comment #2 previously published in the discussion, as a starting point.

1. *The main problems can be summarized as: 1: The language and structure of the text is not good. It would strongly benefit from a detailed English editing and some rewriting of sections to make certain aspects clearer. 2: Although the paper does not focus on the main scientific achievements of Perrault, the authors cannot assume that the reader is fully aware of these. They may have been highlighted in the paper of Nace (1974), however that paper might not be directly available (I had to order it).*

   We reviewed the paper according to the suggestions, please see below in the detailed comments;

2. *I suggest that the authors add the reference to English translation of Perrault's work. This makes it possible for interested readers to trace and read the work itself. La Roque, A. (1967). On the origin of springs, Pierre Perrault (translation). New York and London, Hafner Publishin.g Co.*

   All the citations of Perrault's opus were taken from La Roque's 1967 translation, but in the first version of the paper we chose to cite it as Perrault (1967) and we did not note that the compiler missed the translation note in the references section. In the present version we explicitly recalled the (main) translators directly in the reference citation, so that they are acknowledgeable by the reader. The resulting citation is e.g. Perrault and LaRoque (1967);

3. *the abstract does not represent a proper summary of the main text, currently it reads more like an introduction. It needs to be completely rewritten.*

   The abstract was modified according to the suggestion. Particularly we shortened the first part (first two paragraphs), relieving its introductory style and better focusing on Perrault's opus. Than we better evidenced which are the focus and the original contribution of the paper in the second part.

4. *The introduction should better explain/summarize the main achievements of Perrault as known and described by e.g. Nace (1974). Then is should explain what the issue is that is not known well about Perrault and how this paper will enlighten us in showing these aspects of Perrault's work.*

   We reorganized the Introduction according to the suggestion of firstly recalling Perrault's main achievements and enlightening Nace's (1974) contribution to their comprehension, and then of better detailing our focus. We also enucleated the analysis of the cultural context at Perrault's ages in a standing–alone new Section 2, between the Introduction and the section regarding the comparison between Perrault's and Schott's opus;

5. *The English grammar and spelling needs to be improved. Now there are sentences, which are difficult to understand or where the formulation is imprecise(...)*

   *The style of writing is also with a lot of long sentences and paragraphs, which makes it more difficult to follow. An example is p6, line 23-27. Line 4 page 2, '... in order to become a modern science, Hydrology needed two steps...'; I do not like this style of de-personified subjects. Sciences, including hydrology is advanced by people, crediting then the science itself with steps 'needed' is, especially in a historical oriented paper, to be avoided*

   English grammar and spelling were thoroughly reviewed, and too long sentences were cut in order to make the paper better readable. In the previous version of the paper we capitalized the subjects (Hydrology, Physics, etc.) according to a common usage in the historical and philosophical milieu. We preferred to maintain this usage in a revised manuscript, and we paid attention avoid it when it was misleading, as in the cases raised by the Reviewer. In that cases we rephrased the sentence;

6. *E.g. the very first words of the abstract state: 'The studies conducted...', which studies do the author refer to?*

   We referred to naturalistic and philosophical studies. The sentence was rephrased while reviewing the abstract;

7. *P.6, line 21: Experiences?*: The word was substituted with *experiments*, thanks for the note;

8. *not all symbols of the equations are explained. Should any parts of the paper (text, formulae, figures, tables) be clarified, reduced, combined, or eliminated?*

   We checked all the symbols. Now all of them should be properly recalled.

9. *I did not find the explanations on the experiments (section 4 and 5) and the accompanying figures very clear, I suggest some strong editing of these sections as well.*

   Sections 4, 5 and 6 (actual numbers) were accurately read and partially rephrased in order to avoid unclear statements. We better detailed the captions of the corresponding Figures;

10. *Add reference to La Roque*: Please refer to item 2 in this list.

[revised manuscript text omitted]

---

## Editor Decision (ED1)

[revised manuscript text omitted]

---

## Author Response (AR2)

Dear Professor,

We thank you very much for the comments and the suggestions provided to the reviewed version of our paper.

In this revised version, we accepted all the grammatical and formal suggestions (our corrections are reported in the attached annotated PDF file).

The point-to-point discussion of your comments is also reported in the annotated PDF file, but, for the sake of commodity, the points for which we provided an extended reply are reported in the table below.

We thank you for your attention,
With kind regards,

Stefano Barontini and Matteo Settura

| Page/ line | Comment | Reply |
|---|---|---|
| 4 / 14 | Bacon's Novum Organum (1620) with its discussion of eliminative induction or hypothesis testing by experiment preceded both Descartes and Pascal ? | Bacon's work was certainly well known by Descartes and other philosophers in France, but it is doubtly that it had a direct impact on French scientists. In other words, the reception of Bacon in France was mediated by the Cartesian circles, and in our view it is more likely that Perrault read Descartes and Pascal than Bacon. Moreover, the question of the relationship between Bacon's empiricism and Descartes' rationalism is intricate and would require a far discussion. For these reasons, we decided to limit us to the similarities between Perrault's assertions and those of other French authors of his age, avoiding general claims about Scientific Revolution that would have implied the reference to a large number of authors and positions. (For a more detailed discussion of the link between Bacon and Descartes see Elodie Cassan (ed.), *Bacon and Descartes: the Geneses of Modern Philosophy*, ENS éditions, Lyon 2014). |
| 6 / 34 | so more like Bacon? | It is in our opinion difficult to say whether Perrault's modus operandi was more Baconist or Cartesian. Indeed, both Bacon and Descartes give value to experience, even if they have a different view on the relationship between senses and intellect. In our opinion, it is clear that Perrault directly read and was influenced by Descartes, while he did not seem to directly know Bacon's work. |
| 7 / 19 | But having said what Charles' experiment was (but not the outcome - why not?) you really have to say what this "experimental activity" was before stating the conclusion. Might be better to reorder this paragraph to say what Pierre did to support his conclusion (briefly since you give the detail later), and then add the later work of Mariotte | We reordered the paragraph according to Editor's suggestions. The new paragraph is: "To prove the validity of his objections, Pierre Perrault designed and implemented an experimental activity, the report of which can be found in the subsequent pages (Perrault, 1674, pp.154—160). He concluded that the water alone has no tendency to deeply penetrate the soil nor to spontaneously outcrop to form springs, thus corroborating his theses (Perrault, 1674, p.160 ff). These issues were also discussed few years later by Mariotte, who recognised the importance of the macropores in water percolation, and by Perrault's brother, Charles. |

| | and Charles to make a more logical sequence ( but are they really relevant here) | In fact in 1690 Charles performed an experiment, together with Philippe de la Hire, during which they buried a clay vessel at a certain depth and connected it with a lead pipe to a cellar, in order to verify the water capability to percolate through the soil (Philip et al, 1989). |
| --- | --- | --- |
| 12 / 3 | not sure what you are trying to say here??? Something like:  a consolidation of current understanding | We propose this new version: "...an exercise of analysis of current understanding" |
| 12 / 14 | This is the first time you have used this term ... and of course it has a rather fuzzy meaning in soil hydrology so you should say what you mean by it - or perhaps how it relates to an interpretation of the percolation experiments of Perrault | We introduced this definition: "
[revised manuscript text omitted]

|---|---|---|---|---|---|
| 0 | 4 | 0.33 | 0.09 | 0 | 0.04 |
| 1 | 2 | 2 | 1.76 | 1.69 | 1.72 |
| 4 | 2 | 1.75 | 1.36 | 1.72 | 1.61 |